# A solution for 4-propylguaiacol hydrodeoxygenation without ring saturation

Zihao Zhang [1,5], Qiang Li [2,5], Xiangkun Wu[1], Claire Bourmaud[3], Dionisios G. Vlachos [2,4] ✉, Jeremy Luterbacher [3] ✉, Andras Bodi [1] ✉ & Patrick Hemberger [1] ✉

We investigate solvent effects in the hydrodeoxygenation of 4-propylguaiacol (4PG, 166 amu), a key lignin-derived monomer, over Ru/C catalyst by combined *operando* synchrotron photoelectron photoion coincidence (PEPICO) spectroscopy and molecular dynamics simulations. With and without iso-octane co-feeding, ring-hydrogenated 2-methoxy-4-propylcyclohexanol (172 amu) is the first product, due to the favorable flat adsorption configuration of 4PG on the catalyst surface. In contrast, tetrahydrofuran (THF)—a polar aprotic solvent that is representative of those used for lignin solubilization and upgrading—strongly coordinates to the catalyst surface at the oxygen atom. This induces a local steric hindrance, blocking the flat adsorption of 4PG more effectively, as it needs more Ru sites than the tilted adsorption configuration revealed by molecular dynamics simulations. Therefore, THF suppresses benzene ring hydrogenation, favoring a demethoxylation route that yields 4-propylphenol (136 amu), followed by dehydroxylation to propylbenzene (120 amu). Solvent selection may provide new avenues for controlling catalytic selectivity.

Lignocellulosic biomass, composed of polysaccharides and lignin, holds immense potential as a renewable source for producing fuels and chemicals, offering a sustainable alternative to fossil supplies[1,2]. Lignin, the largest renewable aromatic resource, can be fractionated and then depolymerized into methoxylated phenyl propanoid monomers in organic solvents, typically using hydrogenolysis, oxidative, or acidolysis processes[3,4]. These lignin-derived molecules require additional hydrodeoxygenation to break the C−O bonds for practical applications[5–7]. However, commonly used catalysts, particularly noble metals such as Ru, Pt, and Pd, tend to hydrogenate the aromatic rings first, leading to the formation of cycloalkane derivatives[8–10]. This arises from the favorable π-coordination and planar adsorption configuration of the monomer to the catalyst surface, yielding aromatic ring hydrogenation as the first reaction step[11]. Strategies have been developed to preserve aromaticity, including the introduction of additional Zn sites to tilt the aromatic ring away from the noble metal surface[12] and the use of $Nb_2O_5$[13–17], $TiO_2$[18], or $WO_x$[19] supports with high oxophilicity to lower the $C_{aromatic}$−O bond cleavage activation energy.

Solvents also play a pivotal role in catalysis, influencing both activity and selectivity. Thus, they may be used to tune reaction rates and pathways towards hydrodeoxygenation of biomass-derived resources[20–23]. In addition, because of the need for a solvent to fractionate and depolymerize lignin in various processes, the presence of solvents, or at least residual solvent, is difficult to avoid in the subsequent processing of lignin-derived compounds. However, the mechanism by which solvents affect the reaction pathways remains largely uncharted. In the context of selectively producing lignin-derived platform molecules, lignin conversion and the

[1]Paul Scherrer Institute, Villigen 5232, Switzerland. [2]Catalysis Center for Energy Innovation, University of Delaware, 221 Academy St., Newark, DE 19716, USA. [3]Laboratory of Sustainable and Catalytic Processing, Institute of Chemical Sciences and Engineering, École Polytechnique Fédérale de Lausanne (EPFL), Station 6, Lausanne 1015, Switzerland. [4]Department of Chemical and Biomolecular Engineering, University of Delaware, 150 Academy St., Newark, DE 19716, USA. [5]These authors contributed equally: Zihao Zhang, Qiang Li ✉e-mail: vlachos@udel.edu; jeremy.luterbacher@epfl.ch; andras.bodi@psi.ch; patrick.hemberger@psi.ch

hydrodeoxygenation of lignin-derived monomers, such as 4-propylguaiacol (4PG), have been explored with different solvents. For example, lower hydrodeoxygenation conversion was observed in 1,4-dioxane than in isooctane, which was attributed to competitive adsorption between 4PG and the oxygen-containing solvent[24]. However, similar final product distributions under batch reactor conditions may conceal differences in the reaction mechanism due to the high pressure and long residence time. In 4PG hydrodeoxygenation to propylcyclohexane (PC), ring saturation may be followed by deoxygenation (4PG → 2-methoxyl-4-propylcyclo-hexanol → 4-propylcyclohexanol → PC; black arrows in Fig. 1). Alternatively, deoxygenated aromatic intermediates may form first, which are then hydrogenated and saturated before product analysis (4PG → 4-propylphenol → propylbenzene → PC; pink arrows in Fig. 1). In such cases, batch reactors are ill-suited to reveal mechanistic details. Therefore, *operando* and time-resolved techniques are needed to understand the solvent effect on the mechanism by following intermediates and products in continuous flow. The hydrogenation/deoxygenation reaction sequence in hydrodeoxygenation is of particular interest here, as it guides valorization strategies to produce aromatics.

To address the reaction sequence, we investigated the hydrodeoxygenation of 4PG over a commercial Ru/C catalyst in various solvents. Isooctane and oxygen-containing tetrahydrofuran were chosen as solvents and compared to solvent-free conditions using *operando* synchrotron photoelectron photoion coincidence (PEPICO) spectroscopy in a continuous flow reactor. *Operando* PEPICO spectroscopy allows the detection of short-lived intermediates[25–27] that elude detection in ex situ experiments, as has been amply shown in lignin valorization studies[28–31]. In addition to allowing for temporal analysis, *operando* PEPICO spectroscopy also enables us to monitor these intermediates thanks to molecular beam sampling. Combining this experimental approach with molecular dynamics simulations, we unveil mechanistic insights into the solvent effect on the hydrodeoxygenation pathway of 4PG. This provides a novel, solvent-based approach to tailor the hydrodeoxygenation route of lignin-derived compounds, ultimately enhancing the selectivity toward arenes.

## Results

### *Operando* setup for hydrodeoxygenation of 4-propylguaiacol
The hydrodeoxygenation of 4-propylguaiacol (4PG) was investigated at the VUV beamline of the Swiss Light Source by detecting intermediates and products using *operando* PEPICO spectroscopy, which combines mass spectrometry and photoelectron spectroscopy. The acquired photoion mass-selected threshold photoelectron spectra (ms-TPES) enable the isomer-selective assignment of elusive and stable species in complex reaction systems[32–34]. The PEPICO endstation and reactor setup used in this work are shown in Fig. 2. We introduced $H_2$ and 4PG over the catalyst in three distinct experiments: solvent-free, as well as co-feeding isooctane or tetrahydrofuran (THF, oxygen-containing) as the solvent. A mixture of $H_2$, solvent, and 4PG vapor passed through a heated microreactor containing a commercial Ru/C catalyst bed at ~1.5 bar pressure. Intermediates and products exiting the microreactor

expanded into high vacuum ($10^{-4}$ mbar) and formed a molecular beam (MB), in which reactive collisions are quickly suppressed. After skimming, the MB entered the ionization vacuum chamber ($10^{-6}$ mbar) and was ionized by monochromatic vacuum ultraviolet (VUV) synchrotron light. The photoions and electrons were detected in delayed coincidence, which enabled us to record mass spectra (MS) and photoion mass-selected threshold photoelectron spectra (ms-TPES) of all products and intermediates desorbing from the catalyst surface. This dataset is unique as it allows for the isomer-selective identification of products and elusive intermediates, which are otherwise difficult to detect and assign by alternative analytical techniques.

### Hydrodeoxygenation of 4-propylguaiacol in a solvent-free environment and in isooctane
4PG hydrodeoxygenation (HDO) was first conducted solvent-free. Ru/C catalyst (10 mg) was pretreated in Ar for 30 mins at 200 °C in the microreactor. After cooling the catalyst down to 40 °C, we introduced 4PG vapor in a 10 sccm $H_2$ flow. During the first 1 h (Fig. 3a), no discernible 4PG or HDO product photoionization mass spectral (MS) signal could be observed. Note that hydrogen clearly exited the reactor as observed by the vacuum pressure increase upon turning on the flow, but spectra were recorded below the $H_2$ ionization energy of AIE = 15.426 eV[35]. Only 4PG peak was seen at *m/z* 166 prior to 4 h, indicating the saturation of the Ru/C catalyst with 4PG but without formation of other products (Fig. 3b). After 5–10 h time on stream, three major peaks at *m/z* 172, 142, and 124 began to emerge and gradually intensify. After 10 h at 40 °C, we raised the reaction temperature stepwise to 200 °C (Fig. 3c). At 80 and 120 °C, the product distribution remained like that observed at 40 °C. However, a new peak at *m/z* 126 showed up at 150 °C and became more prominent as the temperature reached 200 °C. Further peaks, such as at *m/z* 98 and 114, also emerged. To identify these newly formed species, we recorded their photoion mass-selected threshold photoelectron spectra (ms-TPES) and calculated the adiabatic ionization energies of the potential spectral carriers (Table S1). The peak at *m/z* 172 could be assigned to 2-methoxyl-4-propylcyclohexanol (2MPC), which is the product of complete benzene ring hydrogenation of 4PG (Fig. 1). The peak at *m/z* 142 is assigned to ionization of neutral 4-propylcyclohexanol (4PC, *m/z* 142), which also yields the peak at *m/z* 124 after dissociative ionization by water loss at the incident VUV energy of 10.5 eV. To provide evidence for this assignment, we recorded reference mass spectra (Fig. S1) and ms-TPES (Fig. 3d) of 4PC. The formation of 4PC is attributed to demethoxylation of 2MPC, which yields methanol as co-product as evidenced by the ms-TPES of *m/z* 32 (Fig. S2a). The peak at *m/z* 126 can be assigned to propylcyclohexane (PC) based on its ms-TPES, formed after dehydroxylation of 4PC (Fig. 3e). Since PC is only formed above 150 °C, dehydroxylation of 4PC to PC exhibits a slightly higher activation energy than 4PG ring saturation and demethoxylation to PC over Ru/C. The peaks at *m/z* 98 and 114 were assigned to methylcyclohexane (MC) and 4-methylcyclohexanol (4MC), respectively (Figs. S2b, 3f), suggesting that removal of a propyl group can also take place at a higher temperature by deethylation.

**Fig. 1 | Possible 4PG reaction routes.** Two reaction pathways for 4-propylguaiacol hydrodeoxygenation to propylcyclohexane.

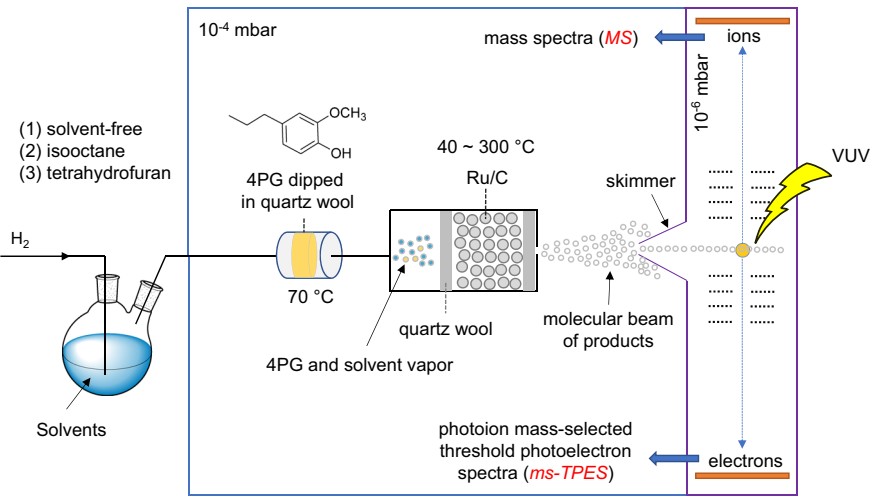

**Fig. 2 | Experimental setup.** Schematic of the *operando* PEPICO spectroscopy setup for 4PG hydrodeoxygenation in various solvent systems.

Next, we investigated 4PG HDO in the presence of isooctane (Fig. S3). Under low reaction temperatures (80 or 100 °C), we observed the same products as in the solvent-free experiment: 2MPC at $m/z$ 172 and 4PC at $m/z$ 142 together with its dissociative ionization fragment at $m/z$ 124 (Fig. S4). As the reaction temperature was increased, PC at $m/z$ 126 emerged as the dehydroxylated product at 150 °C, becoming more pronounced at 200 °C. The signal of isooctane at $m/z$ 99 and 114 coincided with the deethylated MC and 4MC signals (Fig. S4), rendering the identification of the deethylation reaction products in isooctane challenging. As supported by ms-TPES analysis (Fig. S5), we confirm that the reaction pathways are not substantially influenced by the introduction of isooctane, and the 4PG HDO reaction mechanism remains unchanged. In summary, the 4PG HDO mechanism exhibits three regimes both in solvent-free and isooctane experiments depending on the reaction temperature (Fig. 3g): (1) at low temperatures (40–120 °C), the benzene ring is saturated and subsequently demethoxylated; (2) at medium temperatures (150 °C) sequential dehydroxylation takes place; (3) while at high temperatures (>200 °C) a deethylation reaction is observed. All products are ring-saturated, highlighting the previously reported favorable planar adsorption configuration on noble metal catalysts, where the benzene ring lies parallel to the metal surface[9]. This adsorption configuration favors initial benzene ring saturation and makes it nigh impossible to preserve the inherent and valuable aromaticity of lignin in HDO.

### Hydrodeoxygenation of 4-propylguaiacol in tetrahydrofuran
Oxygen-containing solvents bind more strongly to noble metal surfaces than hydrocarbon solvents, possibly due to the oxygen lone pair similar to strong $H_2O$ adsorption on Ru[36], often resulting in suppressed reaction rates due to the competitive binding of the solvent to the active sites[24]. To test whether solvent binding also affects the reaction mechanism, we conducted 4PG HDO experiments using tetrahydrofuran (THF, 72 amu), a solvent also derived from biomass, analogously to the non-solvent and isooctane experiments. Upon introducing $H_2$ flow alongside 4PG and THF vapors at 80 °C, the 4PG signal at $m/z$ 166 appeared already at 20 min time on stream (Fig. 4a), much faster than in the solvent-free (Fig. 3a) and isooctane systems (Fig. S4). This suggests more rapid surface saturation due to substantial coverage of the Ru sites with THF. Notably, no new species form even at extended reaction times at 80 °C (Fig. 4a, b), indicating that THF blocks the active sites and quantitatively inhibits 4PG HDO at this temperature. Upon raising the temperature to 150 °C, major peaks at $m/z$ 136, 124, and 120 were observed (Fig. 4c). Further increasing the temperature to 250 °C did not significantly affect the peak at $m/z$ 124, but peak intensities at $m/z$ 136 and 120 continued to grow. A minor peak could

be identified at $m/z$ 142 and 172 at reaction temperatures of 150 and 200 °C. The $m/z$ 124 peak arose from 4PC dissociative ionization, just as in the non-solvent and isooctane systems (Fig. S6). The lower intensity peaks at $m/z$ 172, 142, and 124 indicated that the predominant 4PG HDO to 4PC route via 2MPC in the solvent-free and isooctane systems is suppressed in the presence of THF. In contrast, the formation of $m/z$ 136 and 120 is more pronounced. Based on their respective ms-TPES (Fig. 4d, e), these two peaks were identified as 4-propylphenol (4PP) and propylbenzene (PB), respectively. This indicates that demethoxylation and dehydroxylation reactions take place prior to benzene ring saturation in the presence of THF, representing a new 4PG HDO reaction pathway (Fig. 4g). Even at temperatures as high as 250 °C, the reactant 4PG signal remained high, indicating lower reaction rates in THF, consistent with previous findings[24].

While the signal at $m/z$ 126 could be assigned to PC based on its ms-TPES (Fig. 4f), the peaks of the aromatic 4-propylphenol (4PP) and propylbenzene (PB) dominate the product distribution (Fig. 4c). Moreover, due to the much stronger peak intensity of 4PP compared to 2MPC, it is anticipated that, instead of 2MPC demethoxylation, 4PC mainly forms by the hydrogenation of 4PP in the presence of THF (Fig. 4g). Additionally, PB is also a feasible precursor for PC. However, due to suppressed benzene ring hydrogenation, much less PC forms even at 250 °C. These results imply that adding an inert solvent coordinating comparatively strongly to the active sites of the catalyst can preserve the aromaticity of 4PG HDO products by altering the reaction pathway at the cost of a reduced reaction rate. As seen in Fig. 4g, the 4PG HDO mechanism in THF can be categorized into two reaction regimes in the 150–200 °C temperature range: (1) the 4PG → 4PP → PB route leads to the formation of PB; while (2) the 4PG → 4PP → 4PC and 4PG → 2MPC → 4PC routes result in the formation of 4PC. Because of the large 4PP abundance, it is also the likely intermediate on the way to 4PC, as opposed to 2MPC like in the non-solvent and isooctane experiments. However, even at the increased reaction temperature of 250 °C, PC formation remains limited.

### Theoretical insights on 4PG and THF adsorption
To gain insights into the effect of THF on 4PG adsorption and the dynamics on the catalyst surface, we employed density-functional theory (DFT) calculations and on-the-fly machine learning force field simulations. A three-layered $p$-(6×6)-Ru(0001) slab model was constructed to represent the Ru/C catalyst. We compared the distinct flat and tilted adsorption configurations of 4PG on Ru(0001) as well as the configurations with broken O−H bond (dissociative adsorption) (Fig. 5a, S7). Notably, the flat adsorption configuration of 4PG was found to be thermodynamically more favorable than the tilted mode,

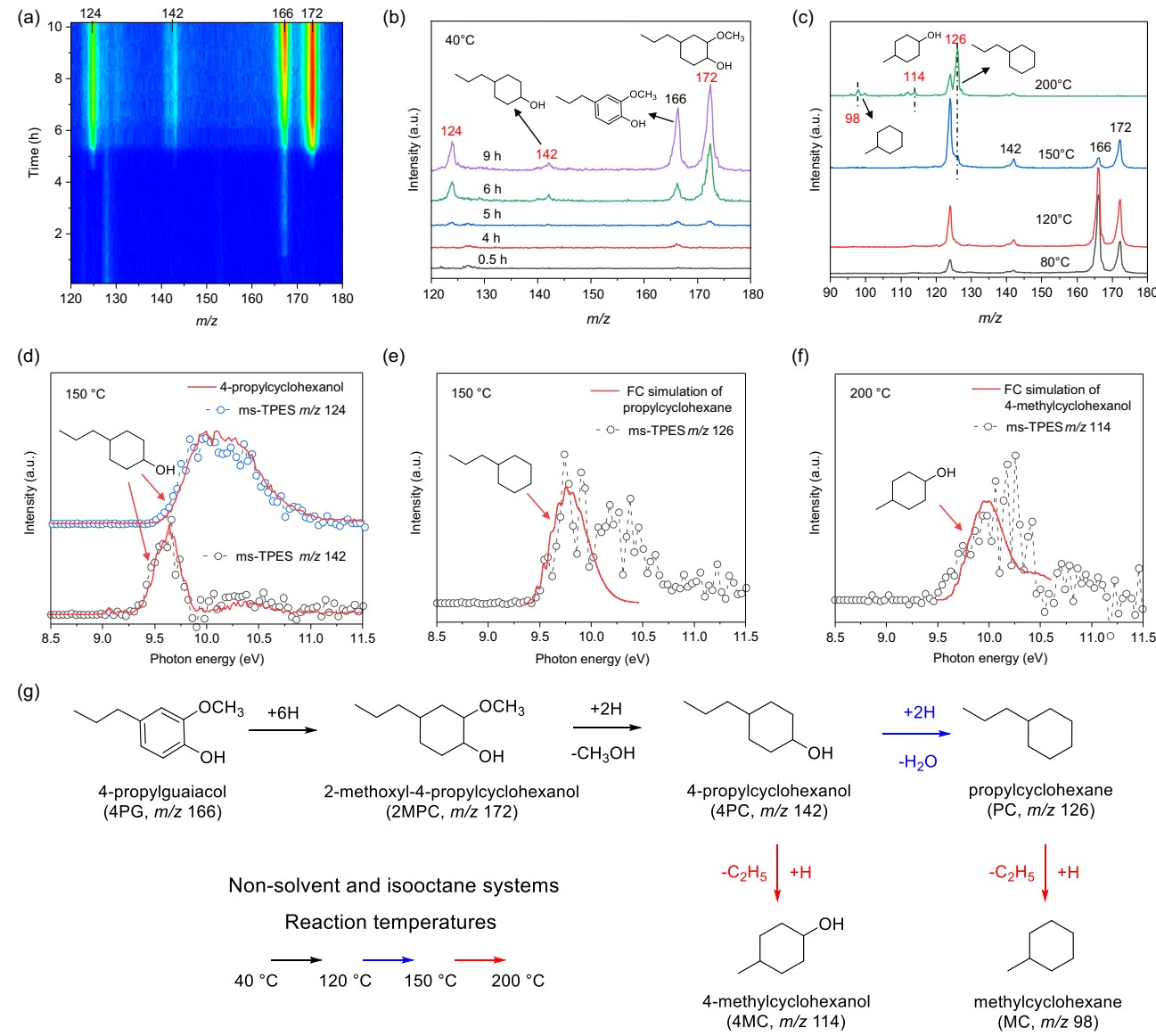

**Fig. 3 | *Operando* PEPICO for 4PG hydrodeoxygenation without solvent co-feed. a** photoionization mass spectra as a function of time on stream at 40 °C. **b** Individual photoionization mass spectra from (**a**) shown for comparison. **c** Photoionization mass spectra at different reaction temperatures; the photon energy used in (**a**–**c**) is 10.5 eV; (**d**–**f**) ms-TPES of (**d**) *m/z* 142, (**e**) *m/z* 126, (**f**) *m/z* 114 at 150 or 200 °C. **g** proposed reaction routes in non-solvent and isooctane systems. Reference spectra and FC simulations are marked by arrows. Reaction conditions: 10 mg of Ru/C, 10 sccm H$_2$, 4PG dipped in quartz wool was maintained at 70 °C, -1.5 bar reactor pressure. (0.04% 4PG, 6% for isooctane, rest H$_2$).

with molecular adsorption energies of −2.54 and −0.84 eV, respectively (Fig. 5a). In addition, dissociative adsorption is exothermic by −3.09 and −1.15 eV in the flat and tilted configurations, respectively, demonstrating the thermodynamic preference for O−H bond scission from 4PG on Ru. The adsorption energy of THF on Ru(0001) was −1.05 eV, comparable to the 4PG adsorption energy of −0.84 eV in the tilted configuration. Furthermore, considering each surface atom as one site, the flat or tilted adsorption configuration of 4PG occupies 10 or 4 sites, while an adsorbed THF molecule occupies 4 Ru sites, respectively. (Fig. 5a). We compared the adsorption energy per site and found that the interaction of THF with the Ru surface is favored over 4PG compared with either the flat or tilted molecular adsorption configuration of 4PG. Only the dissociative flat adsorption configuration of 4PG is comparable to THF with respect to the adsorption energy per site, suggesting that 4PG and THF compete for the Ru adsorption sites. Given the high THF/4PG molar ratio (~ 500) employed herein, Ru surface sites were predominantly covered by THF,

decreasing the density of available sites and suppressing the adsorption of 4PG via the flat configuration. This also aligns with the lower conversion observed at the same reactor temperature when THF was co-fed in the stream. From a mechanistic perspective[37], surface THF must first desorb to provide free Ru sites for 4PG adsorption prior to HDO. This requires three and one THF molecules to desorb to accommodate the flat and tilted 4PG adsorption, respectively (see Fig. 5a). The tilted adsorption conformation of 4PG is therefore more likely to form at high surface coverage because of dynamics. In contrast, the adsorption energy of isooctane per Ru site is only −0.19 eV (Fig. S8), significantly lower than that of 4PG and THF. As a result, the local steric hindrance by isooctane coverage is not expected to be significant, and 4PG readily outcompetes isooctane for the Ru sites. This explains why isooctane does not substantially influence the 4PG HDO mechanism, as the experimental observation shows that the ring-hydrogenated 2MPC (172 amu) is the first product in both the iso-octane and the non-solvent system.

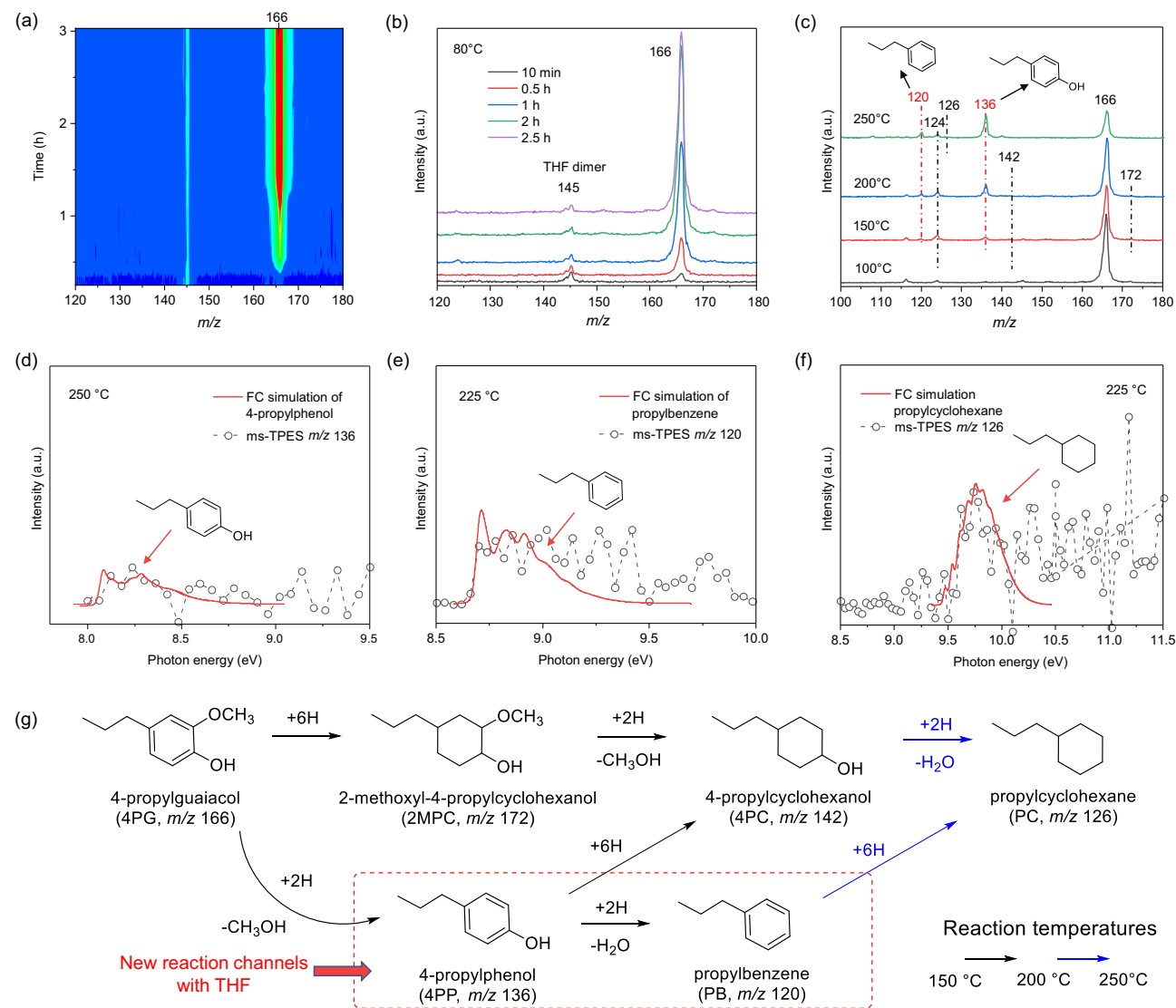

**Fig. 4 | *Operando* PEPICO for the 4PG hydrodeoxygenation in THF.**
**a** photoionization mass spectra as a function of time on stream at a reactor temperature of 80 °C. **b** Selected photoionization mass spectra from (**a**) at different times. **c** Photoionization mass spectra as a function of reactor temperature; the photon energy used in (**a**–**c**) is 10.5 eV; (**d**–**f**) ms-TPES of (**d**) *m/z* 136, (**e**) *m/z* 120, (**f**) *m/z* 126 at 250 or 225 °C. **g** Proposed reaction routes in 4PG HDO in the presence of THF. Franck–Condon simulated spectra are marked by arrows. Reaction conditions: 10 mg of Ru/C, 10 sccm $H_2$, 4PG dipped on quartz wool was maintained at 70 °C, -1.5 bar pressure. (0.04% 4PG, 21% THF, rest $H_2$).

Since the desorption–adsorption equilibrium, the HDO conversion, and selectivity strongly depend on temperature (see Fig. 4c), we employed on-the-fly machine learning force field molecular dynamics (MLMD) simulations, which significantly accelerated the simulation time at the DFT-level of accuracy[38], to investigate the temperature effect on the THF and 4PG adsorption on Ru(0001). To study the adsorption of THF on the Ru catalyst, we performed MLMD simulations from a monolayer THF-covered Ru(0001) surface for 50 ps at three different temperatures (20, 120, and 220 °C), as depicted in Fig. S9. The results revealed that more surface sites became exposed as the temperature increased and THF desorbed, creating free binding sites for 4PG. This temperature-dependent change may explain the absence of 4PG HDO activity at 80 °C (Fig. 4a, b) and why the product formation is only observed at temperatures higher than 150 °C in the presence of THF.

Under the assumption that the flat adsorption of 4PG favors the formation of 2MPC and the tilted adsorption is responsible for the 4PP reaction pathways, we constructed three initial configurations in which we considered 4PG co-adsorption with THF on the surface via the flat and tilted configurations, and 4PG in the gas phase among randomly distributed THF molecules (Fig. S10 at 0 ps). For the MLMD from the 4PG flat adsorption configuration (Fig. 5b, 0 ps), we observed that more THF molecules moved onto the Ru surface at 20 °C (Fig. 5b, 10 ps), but they tended to desorb from the surface at 220 °C (Fig. 5b), in agreement with the MD results of neat THF on Ru without 4PG. In contrast, the flat 4PG remained stable and less affected by THF, suggesting that 4PG in the flat adsorption configuration is indeed stable once formed. The stability of the flat configuration was further assessed using the MLMD for 500 ps at 120 °C starting from the flat adsorption configuration; we observed the O−H scission at 8.4 ps and the persistence of the dissociated flat configuration from 0 to 500 ps (Fig. S11). For the tilted adsorption configuration, 4PG moved away from the Ru surface at 20 °C at 35 ps (Fig. 5c) due to the comparable adsorption energy between tilted 4PG and THF and the latter being present in excess. When the MLMD was run at 220 °C for 10 ps, 4PG desorbed from the Ru surface along with THF, consistent with its lower

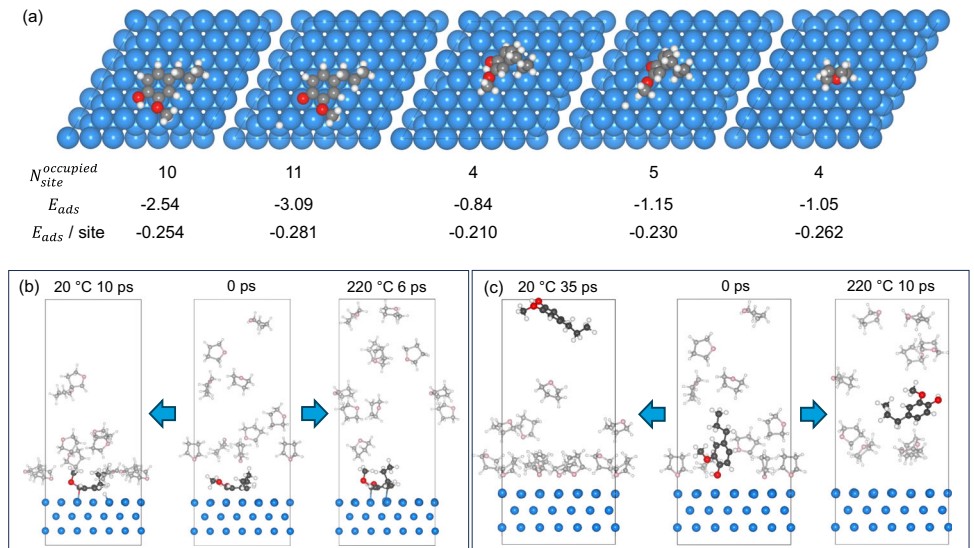

| $N_{site}^{occupied}$ | 10 | 11 | 4 | 5 | 4 |
|---|---|---|---|---|---|
| $E_{ads}$ | -2.54 | -3.09 | -0.84 | -1.15 | -1.05 |
| $E_{ads}$ / site | -0.254 | -0.281 | -0.210 | -0.230 | -0.262 |

**Fig. 5 | Adsorption configurations and MLMD simulations from flat and tilted 4PG configurations. a** Top view of adsorption configurations of 4PG (flat/tilted and molecular/dissociative) and THF on the Ru(0001) site; the occupied Ru sites and adsorption energy (eV) per site for each configuration. **b** MLMD starting from flat adsorption configuration of 4PG in the presence of THF at 20 °C for 10 ps and 220 °C for 6 ps. **c** MLMD starting from tilted adsorption configuration of 4PG in the presence of THF at 20 °C for 35 ps and 220 °C for 10 ps. C and O atoms in THF are in light gray and pink, and in 4PG are in dark gray and red, respectively.

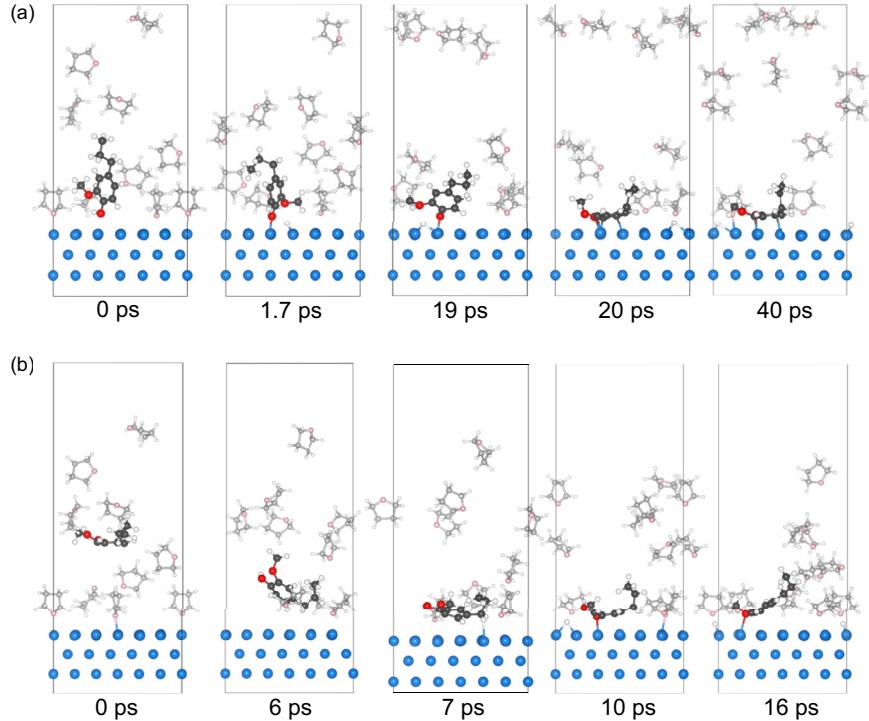

**Fig. 6 | MLMD simulations from tilted and gas phase 4PG configurations.** Evolution of structure changes in MLMD simulation leading to the formation of flat 4PG configuration, starting from (**a**) tilted 4PG configuration at 120 °C and (**b**) gas phase 4PG configuration at 220 °C in the THF atmosphere.

adsorption energy compared to flat configuration (Fig. 5a). For MLMD simulations at a medium temperature of 120 °C (Fig. 6a), the tilted configuration was maintained for 19 ps. However, as more and more THF desorbed from the surface as time progressed, the tilted adsorption configuration eventually transformed into a flat one once the Ru sites for the flat adsorption of 4PG became available. Our results indicate a strong temperature dependence of the competitive 4PG adsorption in the THF atmosphere. At low $T$ (20 °C), the surface as

covered with THF, hindering the 4PG adsorption. However, 4PG in the tilted adsorption configuration was prone to desorb at high temperature (220 °C). At medium $T$ (120 °C), the tilted adsorption configuration of 4PG was stable for a while, but it eventually transformed into a flat configuration once the surrounding Ru sites are liberated from THF.

In our experiment, 4PG was introduced in the gas phase. Therefore, we also investigated this as a starting point for the simulation. We

did not observe 4PG adsorption at 20 °C and 120 °C in the presence of THF (Fig. S12), indicating that 4PG adsorption on the surface is not favored and the reactions are unlikely to occur at low reaction temperatures, consistent with the experimental results shown in Fig. 4b, c. However, as more surface sites are exposed when the temperature increases to 220 °C, 4PG can be adsorbed. The process shows three stages of the evolution of the adsorption geometry, namely from tilted, to flat, and dissociated flat configuration (Fig. 6b). This indicates that 4PG adsorption becomes feasible once the catalyst has sufficient, even if transient, surface sites and it finally leads to the most stable flat adsorption configuration. As the temperature increases, the initially complete surface THF coverage decreases, exposing more and more Ru sites to 4PG adsorption. However, partial THF coverage still creates a steric hindrance to selectively block the flat adsorption configuration, requiring more Ru sites (Fig. 5a). This explains why 4PP and PB dominate the product distribution in Fig. 4c, when THF is present. Nonetheless, the tilted adsorption configuration eventually transitions into the flat one in the MD simulation once more Ru surface is exposed as the temperature increases. Therefore, the experimentally observed change in the HDO mechanism can be attributed to the competition between deoxygenation in the tilted configuration and the conversion of the tilted configuration to the flat one, which entails benzene hydrogenation. Accordingly, abundant $H_2$ may accelerate the conversion of adsorbed 4PG in the tilted configuration before transforming into the flat adsorption configuration. The driving force for the chemical transformation is also that the dissociative adsorption is more exothermic than non-dissociative adsorption (Fig. 5a). The dissociative configuration has broken O−H bond, making C−O(*) bond much more stable than C−O(CH$_3$). Thus, demethoxylation must occur prior to the dehydroxylation, resulting in the formation of 4PP and PB step by step. This also agrees with the reaction sequence that −OCH$_3$ decomposition occurs earlier than C−OH bond scission from 2-methoxy-1,1′-biphenyl[39] and guaiacol[40]. As the temperature increases, the blocking effect of THF becomes weaker, and the transformation from tilted to flat configuration is favored, resulting in a benzene ring adsorption on Ru sites via the d-π interaction, which promotes the further hydrogenation to 4PC and PC through 4PP and PB (Fig. 4c, g).

## Discussion

The combination of *operando* PEPICO and molecular dynamics simulations sheds light on the impact solvent affinity to the catalyst surface has in the 4PG hydrodeoxygenation (HDO) reaction mechanism. In the absence of solvents, the favorable flat adsorption configuration of 4PG on the Ru surface leads to rapid benzene ring saturation, making the formation of 2MPC the first intermediate (left, Fig. 7). Deoxygenation reactions, including demethoxylation and dehydroxylation, only occur after the benzene ring is saturated. Importantly, demethoxylation has a lower energy barrier than dehydroxylation, tentatively ascribed to the stability of the dissociative adsorption configuration, in which the breaking of the O−H bond stabilizes the remaining C−O one. When using isooctane, which exhibits only weak binding to the Ru surface, as a solvent, competitive adsorption between the solvent and 4PG is negligible. Consequently, the reaction mechanism with isooctane resembles the solvent-free system. The weak solvent interaction allows 4PG to preferentially adsorb in the flat configuration, leading to similar reaction pathways and product distributions. In the case of oxygen−containing THF as the solvent (right, Fig. 7), competitive adsorption between THF and 4PG is introduced because of the strong Ru−O (O in THF) interaction. Given an excess of THF compared to 4PG, the Ru/C surface is fully covered by THF at low temperatures (<100 °C), resulting in negligible 4PG conversion (Fig. 4b). As the temperature increases to around 150 °C, the adsorption−desorption equilibrium shifts toward THF desorption (Fig. S9), exposing more Ru surface atoms, which become available for 4PG hydrogenation. Each adsorbed THF molecule occupies approximately 4 Ru sites, similar to the tilted

4PG adsorption configuration (Fig. 5a). In theory, the desorption of a single THF molecule creates enough space for the tilted coordination of a 4PG molecule. However, for the flat 4PG adsorption configuration, which requires 10 Ru sites, the prerequisite is the desorption of at least three adjacent THF molecules from a fully covered surface. Therefore, the presence of THF sterically hinders the thermodynamically favored flat adsorption configuration while allowing only a tilted 4PG adsorption. Consequently, the primary reaction routes in 4PG HDO with THF as the solvent is the demethoxylation reaction to form 4PP, which can subsequently dehydroxylate to form PB (Fig. 7), keeping the aromatic ring intact.

This variation in the reaction pathway demonstrates the significant impact of the solvent on the outcome of the HDO reaction pathway. It suggests that by selecting a solvent with the proper bonding to Ru and controlling the competitive adsorption by temperature and concentrations, new opportunities open up for selective transformations in the hydrodeoxygenation of bio-derived compounds. Additionally, a suitable reactor type is essential for reaction mechanism investigations and to enable the identification of solvent-induced mechanistic changes. Our study suggests that the benzene ring can be saturated over Ru/C catalyst even at room temperature. In batch reactor experiments[24], the high temperatures and pressures applied over a generally long reaction time result exclusively in ring-saturated products, even in the presence of an oxygen-containing solvent. The products agree with our observation, meaning that the reaction chemistry is not fundamentally affected by the pressure and temperature gaps. Instead, in the continuous flow reactor, the reaction time can be finely tuned to form deoxygenated products in the presence of THF and stop the reaction before the aromatic ring is saturated. Thus, mechanistic changes related to, e.g., competitive adsorption, may best be exploited in continuous flow reactors. Overall, the observed solvent effect and reactor configuration used in our study may be extended to other catalytic systems or even serve to guide the development of specific solvent systems or next-generation catalysts, for example, by partial decoration of an inert constituent on the catalyst surface to block specific adsorption configurations selectively.

## Methods

### *Operando* PEPICO experiments

PEPICO experiments were performed at the VUV beamline of the Swiss Light Source using the double imaging Photoelectron Photoion Coincidence (CRF-PEPICO) spectrometer[41]. A scheme of the PEPICO setup used for 4PG HDO is depicted in Fig. 1, and the detailed description of the end station and beamline can be found elsewhere[34]. 10 mg of 5 wt% Ru on active carbon (Ru/C was purchased from Strem Chemicals Inc.; with 6 wt% Ru based on inductively coupled plasma mass spectrometry; the textural property obtained from $N_2$ adsorption at 77 K is summarized in Table S2; TEM image and particle size distribution is shown in Fig. S13) were packed in the stainless steel microreactor (open inlet, 4 mm inner diameter; 8 cm in length; ~ 0.5 cm in catalyst bed length; outlet, 0.05 mm nozzle diameter) held in place with quartz wool at both ends. 4PG dipped in quartz wool was maintained at 5–90 °C by cooling or heating circulating water. Prior to the catalytic test, 4PG was held at 5 °C to decrease its vapor pressure, and the Ru/C catalyst was then pretreated in 10 sccm Ar at 200 °C for 30 min to remove the catalyst's surface moisture. After that, 10 sccm $H_2$ was fed to the microreactor picking up 4PG vapor at 70 °C sample temperature (Fig. 1) for the solvent-free PEPICO experiment. For the experiment with isooctane or THF, the solvents were introduced according to their respective vapor pressure at room temperature and $H_2$ at a flow rate of 10 sccm. Based on the respective vapor pressures, concentrations of 0.04% for 4PG (70 °C), 21% for THF (25 °C), and 6% for isooctane (25 °C) were introduced in the stream. The typical average absolute pressure in the reactor inlet is 1.5 bar. It should be noted that the actual pressure

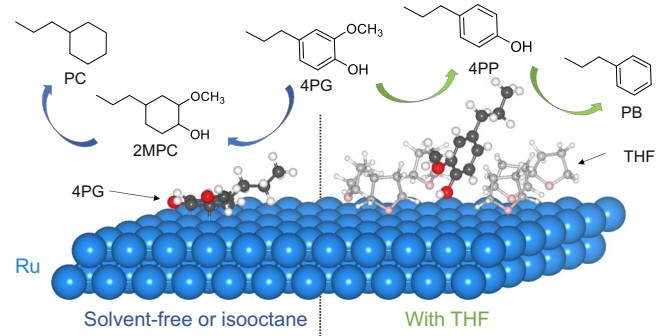

**Fig. 7 | Reaction mechanism.** Schematic reaction mechanism of 4PG HDO over Ru/C with non-solvent, isooctane, and THF systems.

ranges from 1 to 2 bar, depending on the reaction temperatures, and slightly differed with the catalyst filling. After the reaction, the effluent leaves the reactor and forms a molecular beam in the source chamber at a low pressure (~$10^{-5}$ mbar), providing a low-collision environment and preventing the reactive species from being quenched. The molecular beam is skimmed and travels towards the ionization chamber with a lower pressure of $10^{-6}$ mbar, followed by ionization with monochromatic VUV synchrotron radiation. The radiation at higher grating orders is suppressed in the differentially pumped rare gas filter filled with 8 mbar of an Ar, Ne, and Kr mixture. The generated photoions and -electrons are extracted in opposite direction by a constant 216 V cm$^{-1}$ electric field. Both charged particles are detected in velocity map imaging conditions by position-sensitive delay-line anode detectors (Roentdek, DLD40). Photoionization mass spectra are obtained by detecting electrons and ions in delayed coincidence. Photoion mass-selected threshold photoelectron spectra (ms-TPES) are obtained by selecting close-to-zero kinetic energy (threshold) energies and plotting the coincident ion signal in an *m/z* channel as a function of photon energy. Isomers are discerned by comparing ms-TPES with known reference spectra or Franck–Condon simulations and G4 adiabatic ionization energy calculations, as implemented in the Gaussian 16 suite of programs[42].

## Theoretical details

DFT optimization and on-the-fly machine learning force field molecular dynamics simulations were performed using the Vienna ab initio Simulation Package (VASP 6.3.3)[43–46]. The Generalized Gradient Approximation with the Perdew–Burke–Ernzerhof functional (GGA-PBE)[47] was used to obtain the exchange-correlation energies. The Projector Augmented Wave (PAW)[48,49] was chosen to represent the inner electrons, and the cut-off energy of the plane-wave basis set was 450 eV. Gas-phase molecules were calculated in boxes of 20 × 20 × 20 Å$^3$. van der Waals interactions were included via Grimme's DFT-D3 method[50]. The optimized bulk lattice parameters are $a = b = 2.690$ Å ($c/a = 1.582$), close to the experiment values of $a = b = 2.706$ Å ($c/a = 1.582$). The catalyst was modeled by a tri-layer Ru(0001) slab with the bottom layer fixed to mimic the bulk lattice. The Γ-centered *k*-points mesh of 1×1×1 was generated through the Monkhorst–Pack method[51]. A vacuum length of 15 Å was set and dipole correction[52] along the *z*-direction was included. On-the-fly machine learning force fields[45] were employed to perform the ab initio molecular dynamics using the NVT ensemble with a step size of 1 fs and a Nosé–Hoover thermostat. More details for the on-the-fly machine learning are shown in the supplementary information.

## Data availability

The data generated in this study have been deposited in the repository: https://doi.org/10.5281/zenodo.12514988. More detailed data that

support the findings of this study are available from the corresponding author upon request.

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

## Acknowledgements

Experiments were carried out at the VUV beamline of the Swiss Light Source of the Paul Scherrer Institute with support from the PSI CROSS project funding initiative (X.W., A.B., P.H.). This publication was created as part of NCCR Catalysis (Grant No. 180544), a National Centre of Competence in Research funded by the Swiss National Science Foundation (SNSF) (A.B., P.H., Z.Z., C.B., J.L.). The authors also acknowledge funding from the SNSF through grants 200021_182605 (P.H.) and CRSII5_180258 (C.B., J.L.). The work of QL was supported by the Growing Convergence Research (GCR) program at the National Science Foundation (NSF) under award number NSFGCR CMMI. The computations were supported in part through the use of DARWIN computing system: DARWIN—A Resource for Computational and Data-intensive Research at the University of Delaware and in the Delaware Region, which is supported by NSF under Grant Number: 1919839 (D.G.V.). We thank Yu-Cheng Lin for the characterization data on the Ru/C catalyst.

## Author contributions

Z.Z., J.L., A.B., and P.H. conceived the research idea. Z.Z. performed the experiments and analyzed the data. C.B. assisted in experimental design and process. X.W. carried out the FC simulation. Q.L. and D.G.V. carried out the DFT optimization and molecular dynamics simulations. Z.Z., A.B., and P.H. co-wrote and revised the paper, all authors discussed the results and commented on the paper.

## Competing interests

The authors declare no competing interests.
