## [Peer Review File · Nature Communications]

A Solution for 4-Propylguaiacol Hydrodeoxygenation without Ring SaturationREVIEWER COMMENTS

Reviewer #1 (Remarks to the Author):

Zhang et. al. reports the study of mechanistic sequence of hydrogenation vs deoxygenation of 4-propylguaiacol, a lignin monomer, over Ru/C catalyst experimentally using PEPICO spectroscopy which can detect short-lived intermediates and computationally using MD simulations. The reaction was also studied in solvents, non-polar hydrocarbon isooctane and polar aprotic THF, and compared to solvent-free process obtained in PEPICO spectroscopy. It was found that solvent-free and non-polar isooctane solvent as reaction media produce similar products, all with ring saturation, whereas in THF solvent the active Ru sites become occupied with O-atoms of THF and thus reducing benzene ring hydrogenation (even at high temperature of 250°C that 4PG signal persists). Overall, this is a nice piece of work where the combined techniques of experimental spectroscopy and ML-accelerated MD simulations were applied to understand the solvent regulation of hydrodeoxygenation of 4-propylguaiacol (4PG). It may be considered for publication in Nat. Commun. after addressing the following:

1. A couple of reactions involving different chemicals have been described in the first two paragraphs of the introduction. It may be useful for the readers if a scheme can be provided for the reactions/mechanistic possibilities discussed. This is especially needed when discussing the hydrodeoxygenation of 4-propylguaiacol (4PG) (sequence of possible reactions – hydrogenation vs deoxygenation – which one comes first).
2. Not an expert of PEPICO, so personally I find the results discussion on how it works useful and perhaps this will be so for the readers too.
3. The adsorption energy of THF on Ru(0001) is -1.05 eV whereas 4PG in tilted mode is -0.84 eV, this suggests more favourable binding of THF molecule over 4PG in tilted mode. Considering that THF molecules are much more than 4PG molecule, do the authors observe that THF would displace 4PG molecule in tilted mode simulation? As adsorption energy of 4PG in flat mode is -2.54 eV, given long enough equilibration, do the authors observed flat mode binding even with the MD starting from tilted mode as initial configuration (Figure 4c)?
4. For the initial configuration where 4PG is in tilted mode (Figure S10 middle panel), might it be possible that both O atoms of 4PG be coordinated to Ru sites? How would this affect the binding energy of 4PG in tilted mode, as presumably the binding of 2 O atoms from 4PG is more favorable than the binding of 1 O atom?
5. The author claims that “the flat 4PG remained stable and less affected by THF, suggesting that 4PG in the flat adsorption configuration is indeed stable once formed”. However, the simulation was only shown at 10ps for 20°C run and 6ps for 220°C run, does 4PG molecule still stay adsorbed onto the surface in flat mode after a much longer time scale (e.g., 500ps, which has been possible for ML-accelerated ab initio MD, see Chem. Sci., 2023,14, 8338-8354)
6. May the MD simulations be increased in time-scale to see if the conclusions on preferential binding modes/competitive binding between 4PG and THF change?
7. It may be interesting to see how the MD simulations show for isooctane solvent in affecting the binding modes of 4PG upon Ru surface, as the authors recognise the steric effects in affecting the reaction outcome.
8. The following reference on ML-accelerated ab initio MD and explicit solvent effect should be cited: Chem. Sci., 2023,14, 8338-8354.

Reviewer #2 (Remarks to the Author):

This is a fundamental scientific study on unraveling the HDO of propyl guaiacol using a combination of

operando PEPICO spectroscopy experiments and DFT-MLMD simulations. The effects of solvent (isooctane and TGH) on adsorption of the propyl guaiacol molecule and the reaction steps are studied.

Following are my apprehensions on the applicability of the study for a realistic system/process.

It is not clear how the results would serve the understanding of a real system, which involves a mixture of phenolics, furans and linear oxygenates from biochemical constituents. Moreover, propyl guaiacol is not the usual pyrolysate in bio-oil from lignin/biomass. Propyl guaiacol itself would be generated via catalytic hydrogenolysis of lignin. Furthermore, in the case of HDO of biomass oxygenates, vapor phase upgradation is a feasible process that can be scaled-up, while solvent-assisted upgradation is mostly performed in a batch process.

What is the amount of Ru on carbon support, the textural property and the dispersion of Ru in the commercial catalyst?

Therefore, I feel the manuscript is better suited to a hard-core catalysis journal.

Reviewer #3 (Remarks to the Author):

This manuscript examines hydrodeoxygenation of 4-propylguaiacol using a Ru/C catalyst with PEPICO and MD simulations. The study of solvent effects is an exciting application of PEPICO, and definitely of interest to the readership. Overall the paper is very high quality, and I have only a few minor comments/questions.

The reactor pressure in the PEPICO experiment is indicated to be 1-2 bar. Could the authors clarify/confirm that this is an absolute pressure? Can the authors add a statement in the methods section on the basis for writing this pressure range? If it is just from the ideal gas law and reactor/evaporator temperature, it is then helpful to know one well-defined pressure in the system.

In the introduction, when the authors discuss various steps involved in the reaction system (ring hydrogenation, etc) , could they name the primary products produced from hydrodeoxygenation of 4-propylguaiacol, and why some are desired/targeted and some are unwanted/side reactions? I was not familiar enough with this system to understand the motivation.

A commercial 5 wt% Ru/C catalyst is used. I understand catalyst optimization is not the purpose of this study. But there are many different aspects to the catalyst that could be mentioned to help the reader understand the results obtained here compared with those of other Ru-based catalysts. What is the average diameter of particles, is there a surface oxide layer, what is the surface area, what is the stated purity from the manufacturer, etc. If the team does not have access to tools like XPS and TEM, could they reference images/data from other papers that used this source, or from the manufacturer?

Notes: Comments in *blue* - Replies in black - Actions in **bold** - Revision location **highlighted**

Indicated page, figure, or reference numbers refer to the revised Manuscript and/or Supporting Information with changes highlighted.

Reviewer #1

Zhang et. al. reports the study of mechanistic sequence of hydrogenation vs deoxygenation of 4-propylguaiacol, a lignin monomer, over Ru/C catalyst experimentally using PEPICO spectroscopy which can detect short-lived intermediates and computationally using MD simulations. The reaction was also studied in solvents, non-polar hydrocarbon isooctane and polar aprotic THF, and compared to solvent-free process obtained in PEPICO spectroscopy. It was found that solvent-free and non-polar isooctane solvent as reaction media produce similar products, all with ring saturation, whereas in THF solvent the active Ru sites become occupied with O-atoms of THF and thus reducing benzene ring hydrogenation (even at high temperature of 250°C that 4PG signal persists). Overall, this is a nice piece of work where the combined techniques of experimental spectroscopy and ML-accelerated MD simulations were applied to understand the solvent regulation of hydrodeoxygenation of 4-propylguaiacol (4PG). It may be considered for publication in Nat. Commun. after addressing the following:

We thank the reviewer for recognizing the new insights provided by our work. We address their comments point-by-point below.

1) A couple of reactions involving different chemicals have been described in the first two paragraphs of the introduction. It may be useful for the readers if a scheme can be provided for the reactions/mechanistic possibilities discussed. This is especially needed when discussing the hydrodeoxygenation of 4-propylguaiacol (4PG) (sequence of possible reactions – hydrogenation vs deoxygenation – which one comes first).

We thank the reviewer for this insightful suggestion. A new Fig. 1 together with the relevant discussion has been added to **the introduction part**. The included discussion is ‘**In 4PG hydrodeoxygenation to propylcyclohexane (PC), ring saturation may be followed by deoxygenation (4PG → 2-methoxyl-4-propylcyclohexanol → 4-propylcyclohexanol → PC; black arrows in Fig. 1). Alternatively, deoxygenated aromatic intermediates may form first, which are then hydrogenated and saturated before product analysis (4PG → 4-propylphenol → propylbenzene → PC; pink arrows in Fig. 1).**

Fig. 1 | Two reaction pathways for 4-propylguaiacol hydrodeoxygenation to propylcyclohexane.

2) Not an expert of PEPICO, so personally I find the results discussion on how it works useful and perhaps this will be so for the readers too.

We have added a brief introduction about PEPICO spectroscopy **in the section of ‘Operando setup for hydrodeoxygenation of 4-propylguaiacol’**. The included discussion is ‘**...operando PEPICO spectroscopy, which combines mass spectrometry and photoelectron spectroscopy. The acquired photoion mass-selected threshold photoelectron spectra (ms-TPES) enables the isomer-selective assignment of elusive and stable species in complex reaction systems.**³²⁻³⁴ The PEPICO endstation and reactor

setup used in this work are ...' The related references for a more detailed introduction of PEPICO spectroscopy are also cited for the readers who want to know the details about this technique.

3) The adsorption energy of THF on Ru(0001) is -1.05 eV whereas 4PG in tilted mode is -0.84 eV, this suggests more favourable binding of THF molecule over 4PG in tilted mode. Considering that THF molecules are much more than 4PG molecule, do the authors observe that THF would displace 4PG molecule in tilted mode simulation? As adsorption energy of 4PG in flat mode is -2.54 eV, given long enough equilibration, do the authors observed flat mode binding even with the MD starting from tilted mode as initial configuration (Figure 4c)?

Regarding the displacement of 4PG by THF, this process involves two individual steps: 1) the desorption of 4PG and 2) the adsorption of THF. This can be interpreted by Fig. 5C, which depicts the titled 4PG desorbing to the gas phase, followed by the adsorption of THF at 20 °C for 35 ps.

Concerning the configuration evolution from the initial titled configuration, we observed that 1) 4PG desorption in Fig. 5C and 2) 4PG transformation to flat adsorption (as asked by the reviewer) in Fig. 6a. It is important to note that the formation of the flat configuration is initialized by the O—H bond scission, which enhances the interaction between $\text{ArO}^*\text{—Ru}$, thereby stabilizing the dissociated 4PG species on the surface.

4) For the initial configuration where 4PG is in tilted mode (Figure S10 middle panel), might it be possible that both O atoms of 4PG be coordinated to Ru sites? How would this affect the binding energy of 4PG in tilted mode, as presumably the binding of 2 O atoms from 4PG is more favorable than the binding of 1 O atom?

We agree with the reviewer's inference that two Ru—O interactions are more stable than one. This assumption is based on the premise that two Ru—O interactions do not induce other side effects that would raise the energy of the system. For the 4PG adsorption, as depicted in the figure below, we have analysed a configuration featuring two Ru—O interactions (Ru—OCH₃ and Ru—OH). Our results show that this structure is by 0.03 eV slightly less stable than the configuration with one Ru—OH interaction. As a result, **the structure with two Ru—O interactions shows adsorption energy (-0.81 eV) similar to a single Ru—O interaction. This is due to two compensating factors counteracting the increased coordination number: 1) the hydrogen bond between OH and OCH₃ is weakened with respect to a single Ru—O interaction as the length of the hydrogen bond is elongated from 2.018 to 2.172 Å, 2) steric hindrance inhibits the effective interaction between Ru and OCH₃, resulting in a longer Ru—OCH₃ distance (3.035 Å) than in Ru—OH (2.333 Å). We have added the tilted configuration with two Ru-O interactions and the above discussion into Fig. S7 and its caption.**

5) The author claims that “the flat 4PG remained stable and less affected by THF, suggesting that 4PG in the flat adsorption configuration is indeed stable once formed”. However, the simulation was only shown at 10ps for 20°C run and 6ps for 220°C run, does 4PG molecule still stay adsorbed onto the surface in flat mode after a much longer time scale (e.g., 500ps, which has been possible for ML-accelerated ab initio MD, see Chem. Sci., 2023,14, 8338-8354)

We acknowledge the reviewer's concern regarding the short time scale, shown at 10ps for 20°C run and 6ps for 220°C run in Fig. 5, might not be long enough to address the stability of the flat configuration. Another example shown in Fig. 6a at 120 °C, we found that the flat configuration (formed at 20 ps) remains stable until the simulation was halted at 40 ps. Our rationale is based on the that fact that the flat adsorption is way

stronger than either the tilted 4PG or THF adsorption, and the O—H bond scission further stabilizes the 4PG on the surface. Following the reviewer's suggestion, we conducted the Machine Learning Force Field MD at 120 °C over a time scale of 500 ps (as shown in Fig. S11) and observed that the flat configuration remains and confirms its highly stability on the surface as shown in the results below. More specifically, we observed that the O—H bond breaks at 8.4 ps and further strengthen the 4PG—Ru interaction. The relevant discussion related to Fig. S11 was added to the Page 10: The stability of the flat configuration is further verified using the MLMD at a longer time scale of 500 ps at 120 °C starting from the flat adsorption configuration, where we observed the O—H scission at 8.4 ps and the persistence of the flat configuration from 0 to 500 ps (Fig. S11).

Fig. S11 MLMD starting from flat 4PG (circled) in the presence of THF at 120 °C after 500 ps. The inset at 8.4 ps illustrates the local structure of O—H bond dissociation.

6) May the MD simulations be increased in time-scale to see if the conclusions on preferential binding modes/competitive binding between 4PG and THF change?

The MD simulation reveals the dynamics of the surface competitive adsorptions of 4PG and THF. As the binding energy of tilted 4PG and THF are close, we can estimate that the steric hindrance effect that we observed at the time scale of 20-40 ps will exist even if the simulation was conducted over a longer time scale. For the flat 4PG and THF adsorption, we have conducted a MD test for 500 ps as recommended in Comment #5 and our results show that the flat configuration is still stable on the surface, supporting our conclusion that the time scale applied in our MD simulation is adequate.

7) It may be interesting to see how the MD simulations show for isooctane solvent in affecting the binding modes of 4PG upon Ru surface, as the authors recognise the steric effects in affecting the reaction outcome.

In our results, we have demonstrated the impact of THF on the interactions between 4PG and catalyst surface. We believe that different solvents may influence the adsorption configurations. Specifically, regarding the isooctane, given its much lower binding energy with the catalyst (Ru in our case) compared to THF, it can be estimated that the local steric hindrance would be reduced, facilitating the formation of 4PG flat adsorption, and leading to low selectivity to aromatics. **We have included the discussion regarding the isooctane effect in Page 9: the local steric hindrance by isooctane coverage is not expected to be significant, and 4PG readily outcompetes isooctane for the Ru sites. This explains why isooctane does not substantially influence the 4PG HDO mechanism, as shown by the experimental observation that the ring-hydrogenated 2MPC (172 amu) is the first product in both the isooctane and the non-solvent system.**

8) The following reference on ML-accelerated ab initio MD and explicit solvent effect should be cited: Chem. Sci., 2023,14, 8338-8354.

In this recommended work, the authors conducted the ML assisted MD simulations to achieve a rapid prediction of explicit solvent impact on adsorption and reactions in heterogeneous catalysis. It is highly relevant to our research, which focuses on the THF (solvent) effects on controlling the 4PG (reactant) adsorptions. **We have cited this work as reference 38 in page 9.** The added description is ‘..., which significantly accelerating the time scale of MD simulation while maintain the accuracy at the DFT level,³⁸...’

Reviewer #2

This is a fundamental scientific study on unraveling the HDO of propyl guaiacol using a combination of operando PEPICO spectroscopy experiments and DFT-MLMD simulations. The effects of solvent (isooctane and TGH) on adsorption of the propyl guaiacol molecule and the reaction steps are studied. Following are my apprehensions on the applicability of the study for a realistic system/process. We thank the Reviewer for their comments in our paper and have addressed them below.

It is not clear how the results would serve the understanding of a real system, which involves a mixture of phenolics, furans and linear oxygenates from biochemical constituents. Moreover, propyl guaiacol is not the usual pyrolysate in bio-oil from lignin/biomass. Propyl guaiacol itself would be generated via catalytic hydrogenolysis of lignin. Furthermore, in the case of HDO of biomass oxygenates, vapor phase upgradation is a feasible process that can be scaled-up, while solvent-assisted upgradation is mostly performed in a batch process.

We agree with the reviewer that the understanding of solvent effects on lignin-derived bio-oils needs further studies. Like the reviewer said, the composition of bio-oil is quite complex, involves a mixture of phenolics, furans and linear oxygenates, etc. The solvent effect on upgrading of various compounds is different, which makes the mechanism study of solvent effect on real bio-oil highly challenging or even impossible. The valorization of lignin includes two different routes: (1) lignin to bio-oil, followed by bio-oil upgrading and (2) selective catalytic depolymerization of lignin to monomers, such as 4-propylguaiacol, followed by hydrodeoxygenation. Obviously, our goal focuses on route (2) rather than (1). Since solvents are needed in the lignin depolymerization process, solvent-assisted hydrodeoxygenation of 4-propylguaiacol could avoid the costly drying step and thus becomes a promising route. Our goal was to understand the model mechanism by which solvents affect 4-propylguaiacol hydrodeoxygenation. While we agree with the referee that the gap between such a fundamental study and applied process design is large, we believe that a mechanistic understanding of catalysis processes can contribute to and guide targeted process optimization and mechanistic studies complement and improve cook-and-look approaches.

Indeed, solvent-assisted processes are mostly carried out in batch reactors. However, like we state in the introduction part, batch reactors are ill-suited to reveal mechanistic details because the key intermediates would lead to the final products under high pressure and long residence time, thus evading detection (*ACS Sustain. Chem. Eng.* 2019, **7**, 16952-16958). Therefore, this work not only provides a clear mechanistic understanding of solvent effects during 4-propylguaiacol hydrodeoxygenation, but also introduces a possibility of solvent-assisted 4-PG hydrodeoxygenation reaction in continuous reactors. In fact, we are in the process of exploring the solvent-assisted hydrodeoxygenation reaction in large scale in the continuous fix-bed reactor in our laboratory. However, the results are out of the scope of this work.

What is the amount of Ru on carbon support, the textural property and the dispersion of Ru in the commercial catalyst?

The amount of Ru on carbon support and textural property were determined by inductively coupled plasma mass spectrometry (ICP-MS) and N₂-adsorption results, respectively. The measured amount of Ru was 60.55 ug/mg catalyst, which equals to 6% Ru loading on the carbon support, which is slightly higher than the stated 5% by Strem Chemicals, Inc. The textural property is summarized in Table S2, in which Ru/C holds a surface area of 671.8 m²/g and pore volume of 0.46 cm³/g. The average diameter of Ru nanoparticles is ca. 2.4 nm, determined by small cluster analysis in a representative transmission electron microscopy (TEM) image. However, the particle size is nonuniform in the commercial Ru/C catalyst and skewed by a small amount of large Ru particles. Therefore, it is difficult to predict Ru dispersion by TEM image. Fortunately, Ru dispersion of this commercial 5% Ru/C catalysts has been widely studied by chemisorption methods, with an estimated Ru dispersion of ~8% (*Green Chem.*, 2017, **19**, 3252). We did not repeat the measurement for Ru dispersion by chemisorption because this does not affect our conclusion. **We have included the ICP-MS, N₂-adsorption, and TEM results for the commercial Ru/C catalyst in the section Methods and Table S1/ Fig. S15: Ru/C was purchased from Strem Chemicals Inc.; with**

Ru of 6 wt% based on inductively coupled plasma mass spectrometry; the textural property obtained from N₂ adsorption at 77 K is summarized in Table S2; TEM image and particle size distribution is shown in Fig. S13.

Therefore, I feel the manuscript is better suited to a hard-core catalysis journal.

The major novelty of this work is to uncover the solvent effect mechanism on the hydrodeoxygenation of an important lignin-derived compound 4-propylguaiacol by combined PEPICO and MLMD simulations. Although a commercial Ru/C catalyst was used in this work, the focus is not to characterize the catalyst structure and correlate the structure-activity relationship that the hard-core catalysis journals typically focus on. As pointed out by the referee, the results go beyond hard-core catalysis applications, as the mechanism of a dynamic competition between certain solvents and reactants on the surface is shown to be driving the changes in reactivity. Therefore, we would argue that our mechanistic work is well-suited to a more comprehensive journal, such as *Nature Communications* since people outside the field of catalysis may be interested in it, too (similar to another fundamental catalysis mechanism paper some of us published there on the guaiacol catalytic pyrolysis, see DOI: 10.1038/ncomms15946). Additionally, this work has been transferred to *Nature Communications* from the catalysis journal *Nature Catalysis* upon an editorial suggestion.

Reviewer #3

This manuscript examines hydrodeoxygenation of 4-propylguaiacol using a Ru/C catalyst with PEPICO and MD simulations. The study of solvent effects is an exciting application of PEPICO, and definitely of interest to the readership. Overall the paper is very high quality, and I have only a few minor comments/questions.

We thank the Reviewer for such a positive impression on our work. We address their comments/questions point-by-point below.

The reactor pressure in the PEPICO experiment is indicated to be 1-2 bar. Could the authors clarify/confirm that this is an absolute pressure? Can the authors add a statement in the methods section on the basis for writing this pressure range? If it is just from the ideal gas law and reactor/evaporator temperature, it is then helpful to know one well-defined pressure in the system.

The measured reactor pressure in our PEPICO experiment is indeed absolute pressure, which we **have added in the section Methods**. The reaction pressure is measured during the PEPICO experiment. However, the measured pressure is affected by actual reaction temperatures and differences in catalyst filling. We have carefully checked the actual reaction pressures under different conditions, and an average absolute pressure of 1.5 bar was obtained. **We have modified the description in the section Methods as 'The typical average absolute pressure in the reactor inlet is 1.5 bar. It should be noted that the actual pressure ranges from 1 to 2 bar, depending on...'**

In the introduction, when the authors discuss various steps involved in the reaction system (ring hydrogenation, etc), could they name the primary products produced from hydrodeoxygenation of 4-propylguaiacol, and why some are desired/targeted and some are unwanted/side reactions? I was not familiar enough with this system to understand the motivation.

A new Fig. 1 together with the relevant discussion has been added to **the introduction part**. The included discussion is **'In 4PG hydrodeoxygenation to propylcyclohexane (PC), ring saturation may be followed by deoxygenation (4PG → 2-methoxyl-4-propylcyclohexanol → 4-propylcyclohexanol → PC; black arrows in Fig. 1). Alternatively, deoxygenated aromatic intermediates may form first, which are then hydrogenated and saturated before product analysis (4PG → 4-propylphenol → propylbenzene → PC; pink arrows in Fig. 1)'**. The primary products through two various reaction pathways are named in the main text and Fig. 1. In the first paragraph of the Introduction section, we have introduced that commonly used catalysts, particularly noble metals such as Ru, Pt, and Pd, tend to

hydrogenate the aromatic rings first due to the favorable π -coordination and planar adsorption configuration of the monomer like 4PG to the catalyst surface. Strategies have been developed to preserve aromaticity by catalyst modification. Therefore, the desired or value-added products are aromatics, that only exist in the second reaction route in Fig. 1 (pink arrows). The inclusion of Fig. 1 and the description of two different reaction routes will help readers better understand our motivation. We thank the reviewer for this suggestion.

Fig. 1 | Two reaction pathways for 4-propylguaiaicol hydrodeoxygenation to propylcyclohexane.

A commercial 5 wt% Ru/C catalyst is used. I understand catalyst optimization is not the purpose of this study. But there are many different aspects to the catalyst that could be mentioned to help the reader understand the results obtained here compared with those of other Ru-based catalysts. What is the average diameter of particles, is there a surface oxide layer, what is the surface area, what is the stated purity from the manufacturer, etc. If the team does not have access to tools like XPS and TEM, could they references images/data from other papers that used this source, or from the manufacturer?

We thank the reviewer for the suggestion. The average diameter of Ru nanoparticles is about 2.4 nm, determined by small cluster analysis in a representative transmission electron microscopy (TEM) image. Based on the experimental conditions used (H_2 and high temperature), we can assume that the catalyst is reduced *in situ*, and therefore does not have any oxide layer. The textural property was summarized in Table S2 and Figure S13, in which Ru/C hold a surface area of 671.8 m^2/g and pore volume of 0.46 cm^3/g . The amount of Ru on carbon support was determined by inductively coupled plasma mass spectrometry (ICP-MS). Ru amount is 60.55 $\mu g/mg$ catalyst, which equals to 6% Ru loading on carbon support slightly higher than the value (5%) provided by Strem Chemicals, Inc. **We have included the ICP-MS, N_2 -adsorption, and TEM results for the commercial Ru/C catalyst in the section Methods and Table S1/ Fig. S15: Ru/C was purchased from Strem Chemicals Inc.; with Ru of 6 wt% based on inductively coupled plasma mass spectrometry; the textural property obtained from N_2 adsorption at 77 K is summarized in Table S2; TEM image and particle size distribution is shown in Fig. S13.**

REVIEWERS' COMMENTS

Reviewer #1 (Remarks to the Author):

The authors have addressed all the comments thoroughly. The authors should specify the ML algorithm and the detailed calculation setup used for MLMD aspect of the work in the Supporting Information, to ensure that interested readers would be able to replicate the study. This reviewer supports the publication of this manuscript in Nature Communications after the detailed methods for MLMD (including computational software used) is included; no further review from this reviewer is required.

Reviewer #3 (Remarks to the Author):

The authors have adequately responded to reviewer points, and I have no further comments

Manuscript NCOMMS-24-12493-T – Response to Reviewers

Notes: Comments in *blue* - Replies in black - Actions in **bold** - Revision location highlighted

Indicated page, figure, or reference numbers refer to the revised Manuscript and/or Supporting Information with changes highlighted.

Reviewer #1

The authors have addressed all the comments thoroughly. The authors should specify the ML algorithm and the detailed calculation setup used for MLMD aspect of the work in the Supporting Information, to ensure that interested readers would be able to replicate the study. This reviewer supports the publication of this manuscript in Nature Communications after the detailed methods for MLMD (including computational software used) is included; no further review from this reviewer is required.

We thank the reviewer for the additional suggestion. **The ML algorithm and detailed calculation setup for MLMD including used computational software has been included in the Supplementary Information.**

Reviewer #3

The authors have adequately responded to reviewer points, and I have no further comments

We thank the Reviewer for the kind words.